# Structure-Function Relationship of the Ryanodine Receptor Cluster Network in Sinoatrial Node Cells

**DOI:** 10.3390/cells13221885

**Published:** 2024-11-14

**Authors:** Alexander V. Maltsev, Valeria Ventura Subirachs, Oliver Monfredi, Magdalena Juhaszova, Pooja Ajay Warrier, Shardul Rakshit, Syevda Tagirova, Anna V. Maltsev, Michael D. Stern, Edward G. Lakatta, Victor A. Maltsev

**Affiliations:** 1National Institute on Aging, NIH, Baltimore, MD 21224, USA; maltsevav@mail.nih.gov (A.V.M.); shardul123@gmail.com (S.R.); syevda.tagirova@nih.gov (S.T.); sternmi@mail.nih.gov (M.D.S.); lakattae@mail.nih.gov (E.G.L.); 2Department of Cardiovascular Electrophysiology, The Johns Hopkins Hospital, Baltimore, MD 21287, USA; 3School of Mathematics, Queen Mary University of London, London E1 4NS, UK; a.maltsev@qmul.ac.uk

**Keywords:** β adrenergic stimulation, numerical model, pacemaker function, ryanodine receptor, sinoatrial node

## Abstract

The rate of spontaneous action potentials (APs) generated by sinoatrial node cells (SANC) is regulated by local Ca^2+^ release (LCR) from the sarcoplasmic reticulum via Ca^2+^ release channels (ryanodine receptors, RyRs). LCR events propagate and self-organize within the network of RyR clusters (Ca release units, CRUs) via Ca-induced-Ca-release (CICR) that depends on CRU sizes and locations: While larger CRUs generate stronger release signals, the network’s topology governs signal diffusion and propagation. This study used super-resolution structured illumination microscopy to image the 3D network of CRUs in rabbit SANC. The peripheral CRUs formed a spatial mesh, reflecting the cell surface geometry. Two distinct subpopulations of CRUs were identified within each cell, with size distributions conforming to a two-component Gamma mixture model. Furthermore, neighboring CRUs exhibited repulsive behavior. Functional properties of the CRU network were further examined in a novel numerical SANC model developed using our experimental data. Model simulations revealed that heterogeneities in both CRU sizes and locations *facilitate* CICR and increase the AP firing rate in a cooperative manner. However, these heterogeneities reduce the effect of β-adrenergic stimulation in terms of its *relative change* in AP firing rate. The presence of heterogeneities in both sizes and locations allows SANC to reach higher *absolute AP firing rates* during β-adrenergic stimulation. Thus, the CICR facilitation by heterogeneities in CRU sizes and locations regulates and optimizes cardiac pacemaker cell operation under various physiological conditions. Dysfunction of this optimization could be a key factor in heart rate reserve decline in aging and disease.

## 1. Introduction

Sinoatrial node cells (SANCs) located in the posterosuperior right atrium are specialized to generate rhythmic electrical impulses that drive cardiac contractions [1,2]. SANC dysfunction can lead to sick sinus syndrome, which is associated with a variety of life-threatening arrhythmias [3,4,5,6]. The generation of normal rhythmic impulses by SANC is executed via coupled signaling of both cell membrane ion channels and Ca^2+^ cycling, known as the coupled-clock system [7]. A vital component of this system is the sarcoplasmic reticulum (SR), the main intracellular Ca^2+^ store consisting of two parts: (i) the network SR that uptakes cytosolic Ca^2+^ via a Ca^2+^ pump and (ii) the junctional SR (JSR) that embeds clusters of 10–200 Ca^2+^ release channels (ryanodine receptors, RyRs [8]), forming functional couplons or Ca^2+^ release units (CRUs) linked to L-type channels [9]. While RyRs release Ca^2+^ from JSR, their openings are also activated by Ca, creating positive (i.e., explosive) feedback known as Ca-induced-Ca release (CICR) [10]. Thus, RyRs within a CRU tend to release Ca^2+^ in synchrony, creating an elementary Ca^2+^ signal known as a Ca^2+^ spark [11,12]. In SANCs under normal conditions, a Ca^2+^ spark, in turn, can interact with neighboring CRUs via CICR, creating a series of locally propagating sparks observed as Ca^2+^ wavelets, known as local Ca^2+^ release (LCR) (review [7]). The LCRs, in turn, activate an electrogenic Na/Ca exchanger embedded in the cell surface membrane that accelerates diastolic depolarization [7,13,14,15]. The diastolic depolarization is further accelerated by a positive feedback mechanism that includes voltage-activated L-type Ca^2+^ channels which not only depolarize the membrane, but also bring more Ca^2+^ into the cell. Thus, the Ca^2+^ influx brings more “fuel” to further explode CICR and diastolic depolarization, ensuring robust action potential (AP) ignition [16].

In this complex chain of signaling events, CICR plays a critical role, and its efficiency must strongly depend on the spatial organization of CRU network, i.e., the distributions of CRU sizes and locations. What is known thus far about the CRU network in SANC and what is still missing? RyRs are indeed clustered under the cell membrane, as observed in immunofluorescence images of SANCs of various species [17,18]. Early simple numerical models of CRU networks in SANCs considered identical CRUs in a perfect square grid [19,20]. These models demonstrated an important role of CICR in cardiac pacemaking, e.g., the emergence of a Ca^2+^ clock and robust AP firing (via stabilizing diastolic Na/Ca exchanger current [20]). Subsequentially, distributions of CRU sizes and nearest-neighbor distances (NNDs) were measured in 2D in tangential sections of confocal microscopy images and included in a 3D model of SANC [21]. That study raised important questions regarding CRU function, such as “when can a Ca^2+^ spark jump?”, i.e., when CICR propagates among neighboring CRUs. CICR and AP firing emerged in the model only when larger-sized CRUs were mixed with another population of smaller CRUs that provided bridges for CICR propagation, indicating the importance of CRU size heterogeneity. More recent theoretical studies have demonstrated that disorder in CRU locations confers robustness but reduces flexibility in heart pacemaking [22], showing the importance of heterogeneity in CRU distances. This was shown, however, in a SANC model with CRUs of identical sizes and hypothetical CRU repulsion, whereas precise organization of the CRU network in 3D and the functional role of heterogeneity of CRU sizes remain unknown. Thus, it is still not understood how the Ca^2+^ clock and the coupled-clock system emerge from the scale and organization of RyRs and CRUs towards the whole-cell function [23,24]. 

The present study approaches this problem by combining experimental methods and numerical modeling. We imaged the precise organization of the CRU network (both sizes and locations) in 3D in rabbit SANCs by super-resolution structured illumination microscopy (SIM), which provided a greater optical (about twice) resolution compared to confocal imaging [25]. Our results revealed that CRU sizes follow a mixture of two Gamma distributions, suggesting distinct subpopulations of smaller and larger CRUs. Furthermore, NNDs of RyR clusters exhibit a power law distribution at short distances, indicating repulsive interactions among CRUs that would restrain CICR and Ca^2+^ signal propagation. Based on these findings, we developed a new mathematical SANC model featuring CRUs of heterogeneous sizes and locations and performed model simulations with different CRU networks to obtain deeper insights. The simulations demonstrated that the complex and flexible structure of the RyR network, including partial disorder, represents a new pacemaker regulatory mechanism that optimizes cardiac pacemaker cell operation under various physiological conditions. 

## 2. Materials and Methods

### 2.1. Enzymatic Isolation of Individual SANC

SANCs were isolated from male rabbits in accordance with NIH guidelines for the care and use of animals, protocol # 457-LCS-2024, as previously described [26]. New Zealand white rabbits (Charles River Laboratories, Wilmington, MA, USA) weighing 2.8–3.2 kg were anesthetized with sodium pentobarbital (50–90 mg/kg). The hearts were removed quickly and placed in solution containing (in mM): 130 NaCl, 24 NaHCO_3_, 1.2 NaH_2_PO_4_, 1.0 MgCl_2_, 1.8 CaCl_2_, 4.0 KCl, and 5.6 glucose equilibrated with 95% O_2_/5% CO_2_ (pH 7.4 at 35 °C). The SAN region was cut into small strips (~1.0 mm wide) perpendicular to the crista terminalis and excised. The final SA node preparation, which consisted of SA node strips attached to the small portion of crista terminalis, was washed twice in nominally Ca^2+^-free solution containing (in mM) 140 NaCl, 5.4 KCl, 0.5 MgCl2, 0.33 NaH_2_PO_4_, 5 HEPES, and 5.5 glucose (pH = 6.9) and incubated on a shaker at 35 °C for 30 min in the same solution with the addition of elastase type IV (0.6 mg/mL; Sigma, Chemical Co., St. Louis, MO, USA), collagenase type 2 (0.8 mg/mL; Worthington, NJ, USA), Protease XIV (0.12 mg/mL; Sigma, Chemical Co.), and 0.1% bovine serum albumin (Sigma, Chemical Co.). The SA node preparation was next placed in modified “Kraftbruhe” solution, containing (in mM) 70 potassium glutamate, 30 KCl, 10 KH_2_PO_4_, 1 MgCl_2_, 20 taurine, 10 glucose, 0.3 EGTA, and 10 HEPES (titrated to pH 7.4 with KOH), and kept at 4 °C for 1 h in KB solution containing 50 mg/mL polyvinylpyrrolidone (PVP, Sigma, Chemical Co.). Finally, cells were dispersed from the SA node preparation by gentle pipetting in the “Kraftbruhe” solution and stored at 4 °C.

### 2.2. Immunolabeling of RyRs

Isolated cells were fixed with 2% formaldehyde in phosphate-buffered saline (PBS, Sigma, St. Louis, MO) and then permeabilized with 0.5–1% Triton X-100/PBS. Nonspecific cross-reactivity was blocked by incubating the samples for 4 h in 1% BSA/PBS (Jackson ImmunoResearch, West Grove, PA, USA). The cells were then incubated with primary anti-RyR antibodies (IgG1, clone C3-33, Afinity BioReagents, Golden, CO, or AP20325PU-N (Origene, Rockville, MD, USA)); for details, see our database at https://doi.org/10.7910/DVN/N0OLGI, (accessed 13 November 2024). The antibodies were diluted in 1% BSA/PBS overnight, washed in PBS, and then incubated with appropriate fluorescence label-conjugated secondary antibodies (Jackson ImmunoResearch, West Grove, PA, USA). 

### 2.3. Structured Illumination Microscopy (SIM)

SIM uses a sharply patterned light source to improve spatial resolution in fluorescence microscopy [27]. SIM can be used to study the structure and function of cells in great detail, as it can provide images with higher resolutions than those of traditional microscopy techniques. It is particularly useful for studying cells and tissues that have fine structural features, such as the SANC, as we report here. Experiments with RyR detection in fixed cells were performed in the experimental instant SIM setup in Hari Shroff’s laboratory at the National Institute of Biomedical Imaging and Bioengineering, as described elsewhere [27,28,29]. The instant SIM is an implementation of SIM well-suited for high-speed imaging, as the images are processed optically rather than computationally to improve the resolution ~1.4×. SANCs were imaged via an oil ×100, 1.49 NA objective and 488 nm excitation; in-house software controlled the optimal laser power and positioning (sectioning) of cell preparation. The system allowed for imaging at resolutions as low as 145 nm laterally and 320 nm axially. Three-dimensional sectioning was performed with a pixel size of 55.5 × 55.5 nm in xy and z stacking of 150 nm. 

### 2.4. Segmentation of Peripheral RyR Clusters in 3D

Our dataset consisted of 31 rabbit SANCs, which included an aggregated total of 70,982 RyR clusters. We present a novel algorithm for high-throughput segmentation of peripheral ryanodine receptor (RyR) clusters in 3D SANCs (Figure 1). The preprocessing procedure included the following steps in order: a 3D median filter to remove salt-and-pepper noise, signal normalization to standardize the intensity range, and finally, 3D contrast-limited adaptive histogram equalization (CLAHE) [30] to enhance local contrast while preserving overall image structure and global contrast. The segmentation was then performed using the 3D StarDist neural network [31], which directly predicts star-convex polyhedra representations for each RyR cluster without relying on the watershed algorithm. The 3D StarDist network learns to separate partially overlapping or contacting RyR clusters by predicting appropriate polyhedra shapes based on the trained model. The ground truth data used for training 3D StarDist were initially generated using Squassh 1.0.24 [32], an open-source software algorithm in ImageJ Fiji Release Version 1.54f, which is part of the MOSAIC suite that utilizes globally optimal detection and segmentation methodologies [33] and incorporates corrections for the microscope’s point spread function [34,35]. The Squassh segmentation results were then fine-tuned using the Adaptive Watershed tool in ORS Dragonfly (2024.1) [36] and further refined by manual editing in ORS Dragonfly to ensure accurate separation of RyR clusters (Figure 2). The final trained model for 3D StarDist achieved an Intersection over Union (IoU) score of 0.68.

To further refine the segmentation of RyR clusters and remove false positives after 3D StarDist segmentation, DBSCAN (Density-Based Spatial Clustering of Applications with Noise) clustering was used on the centroid data of the segmented RyR clusters. DBSCAN is a density-based clustering algorithm that groups together points that are closely packed, marking points that are in low-density regions as outliers. The key parameters for DBSCAN were ϵ and *min_samples*. ϵ determines the maximum distance between two points for them to be considered as part of the same cluster, or in our case, the imaged SANCs. ϵ was set using the median absolute deviation (MAD) method with a scaling factor *c*.
ϵ=median(A)+3·c·MAD

By using the scaled MAD and the median, the ϵ remained robust against the influence of extreme values:MAD=c·median(|A−median(A)|)
where A represents the set of nearest-neighbor distances. Furthermore, the scaling factor c was defined as:$$c=\frac{1}{\sqrt{2}\cdot erfcinv\left(\frac{3}{2}\right)}\approx 1.4826$$
where *erfcinv* is the inverse complementary error function. This adaptation ensures that the scaled MAD approximates the standard deviation when the data follow a normal distribution. The second parameter, *min_samples,* is the minimum number of points required to form a dense region, and it was set to 3. If a point has at least *min_samples* within its ϵ-neighborhood (including itself), it is considered a core point. Points that are not core points but are within the ϵ-neighborhood of a core point are called border points. Points that are neither core points nor border points are classified as noise or outliers.

The largest cluster deduced by DBSCAN (which is the cell itself) was further refined by computing its alpha shape, which captured the peripheral RyR clusters. The optimal alpha value was determined by iterating through a range of values and selecting the one that maximized the density of points on the surface of the alpha shape. Finally, the algorithm generated various data and visualizations, including the alpha shape, surface area, volume, centroid data, volume data, voxel lists, cluster labels, adjacency matrix, and nearest-neighbor distances, for further analysis. An additional outlier test was performed by taking Tukey’s Fences where extreme nearest-neighbor values are filtered out by keeping only the interval:$$\left[Q_{1}-3.0\cdot \left(Q_{3}-Q_{1}\right),\;Q_{3}+3.0\cdot \left(Q_{3}-Q_{1}\right)\right]$$

The visualizations for all 31 cells are presented in Figure 3, and mouse-interactive HTML files constructed with *plotly* Python library are published in GitHub https://github.com/alexmaltsev/SANC/tree/main/3D%20Visualizations (accessed 13 November 2024). Each cell can be individually opened and rotated (with zoom-in or -out) by the reader. We also published our code to use 3D StarDist and all code for data application and processing in GitHub. See the Code Availability section for more details.

## 3. Results

### 3.1. Two Subpopulations of RyR Cluster Sizes

The distributions of RyR cluster sizes in individual SANC were analyzed, with the results shown in Table 1. 

On average, we detected 2285 clusters per cell, and the overall mean of RyR cluster sizes was 0.18 μm^3^ with a standard deviation of 0.13 μm^3^. The distributions were plotted as histograms with 50 bins, and the existence of two right-side-skewed subpopulations was visually apparent, with four representative cells shown in Figure 4. Thus, a Gamma Mixture Model (GMM) was used to model this distribution. A Gamma distribution is often used to model distributions that are asymmetric, with a longer tail on the right side. A GMM is a probabilistic model that assumes the data are generated from a mixture of several Gamma distributions, in our case two distributions. Each component in the mixture is a Gamma distribution with its own parameters, and the mixture is weighted by coefficients that sum to 1. The PDF for a GMM with two components in our case is:Г(α)=(α−1)!
$$p\left(x\right)=\pi \frac{\beta_{1}^{\alpha_{1}}}{Г\left(\alpha_{1}\right)}x^{\alpha_{1}-1}e^{-\beta_{1}x}+\left(1-\pi \right)\frac{\beta_{2}^{\alpha_{2}}}{Г\left(\alpha_{2}\right)}x^{\alpha_{2}-1}e^{-\beta_{2}x}$$
where Г(α) is the Gamma function, *π* is the mixture weight, and α, β are the shape and scale parameters for their respective Gamma distribution components. The GMM was fit to the data using the *scikit-learn* library in Python 3.14, which implements the expectation maximization algorithm to estimate the parameters of the mixture model. The number of Gamma components in the mixture was determined by evaluating models with different numbers of components and selecting the one with the lowest Bayesian Information Criterion to balance goodness of fit with model complexity.

To initialize the GMM, an estimate of the peak locations in the data was obtained using the method of moments for initial parameter estimation, which splits the sorted data at the median and uses the statistics of each half to approximate the parameters of two gamma distributions. The two most prominent peaks were identified and used as initial estimates for the means of the Gamma components. The optimization of the GMM parameters was performed using the minimize function from the *scipy.optimize* module, with bounds set on the parameters to ensure they remain positive and the mixing coefficient remaining between 0 and 1. The GMM analysis revealed that the distribution of RyR cluster sizes in the rabbit sinoatrial node cells is best described by a mixture of two Gamma components, with the results for individual cells shown in Table 2. 

After application of Tukey’s Fences on the sizes, the RyR clusters were aggregated into a set of 70,849 clusters. Next, a histogram with 100 bins was created and a GMM was fit on top of this aggregated set. The fitted parameters of the GMM were as follows in Figure 5: α₁ = 4.35, β₁ = 125, α₂ = 2.22, β₂ = 10.64, π = 0.16. The first gamma component had an average of 0.0333 μm^3^ and a standard deviation of 0.0160 μm^3^, while the second gamma component had an average of 0.2083 μm^3^ and a standard deviation of 0.1398 μm^3^. These results provide a comprehensive characterization of the RyR cluster size distribution in the rabbit SANC, highlighting the presence of two distinct subpopulations with different average sizes and variability. The kurtosis values of the first and second Gamma components were calculated to be 1.38 and 2.70, respectively. The higher kurtosis of the second component suggests that it represents a subpopulation of larger RyR clusters with a heavier tail in the distribution. The 95th percentile of the second Gamma component was found to be 0.478 μm^3^, which was used as a threshold to define the tail of the distribution representing the largest RyR clusters used in further analysis. 

### 3.2. Repulsive Behavior of RyR Clusters in Spatial Distribution

To investigate the spatial arrangement of RyR clusters, the aggregated distances between nearest-neighbor clusters were analyzed. The distribution of nearest-neighbor distances provides insights into potential spatial interactions, such as repulsion or attraction, among the clusters. The overall mean distance between nearest-neighbor RyR clusters was found to be 638.62 nm, with a standard deviation of 208.14 nm (Table 1). Histograms of the nearest-neighbor distances for each individual SANC were constructed using 50 bins and a power law function which was fit to the ascending part of the distribution up to the inflection point, determined as the bin where the histogram values first fell below half of the peak value. 

The power law function used to model the nearest-neighbor distances is given by $p\left(x\right)=Ax^{b}$, where p(x) is the probability of observing a nearest-neighbor distance, A is a normalization constant, and *b* is the power law exponent. Transforming the dataset into a log-log plot, a linear relationship is obtained between log(p(x)) and log(x). This transformation allows for a more straightforward analysis of the power law behavior. The resulting equation in log-log form is: log(p(x))=log(A)+b·log(x). The variable *b* characterizes the spatial distribution of the clusters, with *b* ≈ 1 indicating a random Poisson process, *b* > 1 suggesting repulsion, and *b* < 1 indicating clustering. The power law function is fit to the ascending part of the histogram until the inflection. Four example cells and their associated analysis are shown in Figure 4. The results in Figure 6B and Table 2 show that each cell exhibited repulsion with *b* > 1. Figure 7 is a visual representation of the spatial distribution of RyR clusters, showing varying levels of cluster density on the surface mesh on sample cell 14. Higher-density areas are colored in blue → green → yellow, representing RyR cluster “hotspot zones”, while low-density areas are in purple, exhibiting strong repulsive forces. 

After application of Tukey’s Fences on the distances, the aggregated set of 70,887 nearest neighbor distances was also analyzed, and the results of the analysis revealed a power law exponent of b = 5.15, also indicating a strong repulsion among the RyR clusters at short distances (Figure 6A). 

### 3.3. Structure–Function Relationship of CRU Network in Numerical Simulations of SANC Function 

We simulated AP firing in numerical SANC models (Figure A1) with different distributions of sizes and locations. With respect to sizes, we tested and compared two distributions: one was the realistic (i.e., heterogeneous) distribution obtained in SIM data adopted to the SANC model (re-scaled and re-binned) and the other was the distribution featuring identical CRU sizes, with each CRU size fixed to 48 RyRs representing the average CRU size in the realistic distribution. It is also important to note that each cell model had the same total number of RyRs. With respect to CRU locations, we tested three CRU distributions of NNDs: (i) uniformly random, (ii) realistic, and (iii) crystal-like grid (Figure A2). The distributions were generated by our CRU repulsion algorithm that dynamically changed positions of CRUs from uniformly random locations with no repulsion towards a crystal-like structure with the strongest repulsion. The change was performed in 100 repulsion steps (see details in Appendix B and Appendix A). The realistic NND distribution was found at an intermediate repulsion step with the repulsion, yielding an average NND value closely matching that measured experimentally. 

Finally, each model was tested in basal-state AP firing and in the presence of β-adrenergic receptor (βAR) stimulation. Thus, we tested the operation of six different SANC models under two conditions, resulting in 12 simulations altogether. The results of these simulations are summarized in Figure 8. The simulations (see examples in Appendix A) showed that, for each CRU location distribution type (random, realistic, crystal), the models with realistic CRU sizes had faster AP firing rates vs. those with identical CRU sizes. At the same time, the absolute values of AP firing rates were the largest (reflecting shortest cycle length) in cells with random CRU locations (Figure 8A), but the smallest (reflecting longest cycle length) in crystal-like grids with the strongest CRU repulsion (Figure 8C). The scenario with realistic locations (Figure 8B) had an intermediate AP firing rate of 157 bpm, close to the experimentally reported rates of isolated rabbit SANCs. 

With respect to βAR stimulation, its effect in terms of *relative* change of AP firing CL (or rate) increased as the order of CRU sizes and position increased, i.e., heterogeneity or disorder decreased and reached its maximum in the model with identical CRU located in a crystal-like grid with the least heterogeneity and maximum order. See panel D, with blue and magenta bars in the “Crystal section”, 459 vs. 296 ms, yielding a 55% increase in AP firing rate from 131 vs. 202 bpm. And vice versa, the βAR stimulation effect decreased as the disorder of CRU sizes and position increased (i.e., order decreased) and reached its minimum in the model with realistic CRU located in (uniformly) random locations. See Figure 8D, with orange and red bars in the “Random” section, 311 vs. 255 ms, yielding only a 22% increase in AP firing rate from 192 vs. 225 bpm. On the other hand, the disorder (heterogeneity) of CRU sizes and locations allowed the models to reach higher absolute values of AP firing rates (shorter cycle lengths) both in the basal state and during βAR stimulation. As a result, the highest beating rate of 225 bpm among all 12 tested scenarios was achieved in the SANC model with the highest degree of disorder, representing a cooperative effect of the most heterogeneous (realistic) sizes and uniformly random locations of their CRUs (red diamonds in panel C). The model with realistic CRU sizes and locations demonstrated a 39% βAR increase in AP firing rate (377 vs. 271 ms of CL, or 159 vs. 220 bpm). 

Next, we analyzed our simulations to obtain insights into the biophysical mechanism of the contribution of the heterogenous CRU network to regulating pacemaker function. According to our previous studies, the critical part of the diastolic depolarization is the AP ignition [16] at the threshold of L-type Ca^2+^ current (I_CaL_) activation. In our model, each CRU size is given by the number of RyRs embedded in that CRU. Therefore, in all 12 simulation scenarios, we determined the total number of RyRs in all firing CRUs at −45 mV, i.e., at the threshold of *I_CaL_* activation (Cav1.3 current component in our model of I_CaL_). We found that the AP cycle length in the basal state and during βAR stimulation was linked to the number of firing RyRs (Figure 9), reflecting the degree of order and disorder (heterogeneity) of CRUs and their ability to self-organize to fire earlier and synchronously within the AP cycle. 

## 4. Discussion

### 4.1. Result Summary: Major Findings

The present study, for the first time, examined the detailed structure of the RyR network under the cell membrane in SANCs (Figure 3) at the nanoscale level via SIM super-resolution microscopy. Previous lower-resolution confocal microscopy studies have shown that RyR appears in clusters which are mainly located under the cell membrane in central (primary) SANCs [17]. It was also shown that RyR-mediated LCR events play a crucial role in the coupled-clock system that ensures robust and flexible pacemaking [7]. Many important properties of the network, however, remained unknown. The major findings of the present study include:(1)Two distinct RyR cluster subpopulations with sizes adhering to a Gamma mixture distribution.(2)Notable repulsion between neighboring RyR clusters.(3)Model simulations showed that heterogeneities in both CRU sizes and locations facilitate CICR and increase AP firing rate in a cooperative manner, but decrease the effects of βAR stimulation. However, the highest possible absolute rate during βAR stimulation is reached with heterogeneities in both CRU sizes and locations.

### 4.2. New Image Analysis Methods and Numerical Models

We developed a novel, high-throughput algorithm for the segmentation of peripheral RyR clusters in 3D in SANCs, enabling the analysis of over 70,000 RyR clusters across 31 rabbit SANCs (Figure 1). Our advanced image preprocessing techniques included 3D median filtering, signal normalization, and CLAHE [30]. It enhanced the local contrast while preserving the global image structure, facilitating more accurate segmentation. For the first time in SANCs, the 3D StarDist neural network [31] was trained for the segmentation of RyR clusters and accurately predicted star-convex polyhedra representations for each cluster, improving upon traditional methods that rely on watershed algorithms and often struggle with overlap. The ground truth data for training the neural network were prepared using Squassh [32] and fine-tuned with the Adaptive Watershed tool in ORS Dragonfly [36]. Further refinement through DBSCAN clustering removed false positives. To accurately capture the peripheral RyR clusters directly interacting with cell membrane ion channels and transporters critical for heart rate regulation, the cell periphery was isolated using an alpha shape algorithm, constructing a minimal bounding surface that closely conforms to the shape of the data points, effectively representing the cell’s geometry.

We also modified our previous 3D numerical model of SANC [22] by including new formulations for CRU function (see Appendix B for details). Recent theoretical studies [37,38,39] showed that Ca^2+^ release activation in a CRU (i.e., Ca^2+^ spark activation) depends substantially on the CRU size, i.e., the number of RyRs in a CRU. Thus, our new formulations were created to reflect actual distributions obtained by segmentation of SIM images: they generate repulsion between the CRUs (Figure A2, Appendix A) and CRUs of various sizes (Figure A3). Using the new formulations, we created and tested a series of models with real and extreme settings that revealed importance of heterogeneities in CRU sizes and locations and their joint regulation of CICR to ensure robust and flexible pacemaker function.

### 4.3. Structure–Function Relations of RyR Network in SANC

While it is known that LCRs play an important role in pacemaker function [7], the detailed mechanisms regarding how LCRs emerge from the RyR network remain mainly unknown. Because the RyR network operates via CICR, its properties must critically depend on both CRU *sizes* (generating stronger releases) and *distances,* which define whether the release of each CRU can reach and activate its neighbors (i.e., “when can a Ca^2+^ spark jump?” [21]). Our previous theoretical studies of CRU function in SANCs first showed that synchronization of stochastic CRUs creates a rhythmic Ca^2+^ clock in SANCs via repetitive phase-like transitions within a perfect rectangular grid of CRU networks [19]. Further linking this model with the membrane clock revealed that RyR-Na/Ca exchanger (NCX)-SERCA local crosstalk ensures pacemaker cell function at rest and during the fight-or-flight reflex [20]. 

While previous studies have tested the effects of CRU sizes and distances in separation, here, for the first time, we found that they act together to regulate pacemaking (Figure 8). The CRUs form a functional network in which Ca^2+^ release events, which self-organize (via CICR) into local Ca^2+^ oscillators that create a rhythmic diastolic Ca^2+^ signal known as the “Ca clock”. This, together with the “membrane clock”, forms a coupled-clock system that ensures robust and flexible pacemaking [7]. The diastolic Ca^2+^ signal acts not alone, but within a strong positive feedback mechanism (known as AP ignition [16]) with NCX, membrane potential, and *I_CaL_*. The self-organized Ca^2+^ signal formed by CRUs at the threshold of *I_CaL_* (at the ignition point) must play a critical role in the ignition and, ultimately, AP firing rate regulation. 

Our simulations showed that this critical signal at the threshold of *I_CaL_* is tightly linked to the structure of the RyR network. The heterogeneity in both CRU sizes and distances strongly increased the CICR and the AP firing rate both in the basal state and during βAR stimulation (Figure 9). Thus, to reach the maximal relative increase in firing rate and, at the same time, the maximum absolute rate, the CRU network can transform to increase its heterogeneity, i.e., to increase the contribution of larger RyR clusters. Recent dSTORM studies in ventricular myocytes showed that the RyR clusters can expand and coalesce after application of isoproterenol [40]. Future dSTORM studies will show whether this also happens in SANCs, allowing the RyR network to transform itself to reach maximum performance in fight-or-flight reflex.

We have previously shown [22] that disorder in CRU locations increases CICR due to Poisson clumping characterized by empty spaces and clusters. The CRU clustering makes it easier for released Ca^2+^ to reach a neighboring CRU. But why does a CRU network with heterogeneous CRU sizes also enhance CICR and AP firing rate? Larger CRUs are naturally present in such network (Figure 4 and Figure 5, beyond the vertical line). These larger CRUs have a greater probability and capacity for Ca^2+^ release [39], acting as central hubs that initiate and amplify CICR to the surrounding smaller CRUs. This multi-scale spatial organization enhances the likelihood of CICR propagation from large clusters to smaller ones, promoting synchronization necessary for effective Ca^2+^ signaling. 

Pilot dSTORM studies of RyR clusters in rat SANCs (abstracts [41,42]) fully support the results of the present study. It was found that the cluster sizes vary from 10 to 150 RyRs, and they are organized in couplons, i.e., in apposition to L-type Ca^2+^ channels found largely at the outer membrane in SAN cells. Heterogenous RyR2 clusters were also found, often with small inter-cluster distances, which is indicative of inter-cluster RyR2 activation.

### 4.4. Limitations and Future Studies

Our cell database was limited to 31 cells, and our model simulations were limited to 12 basic scenarios. Future studies in a larger number of cells and with even stronger imaging resolution, like dSTORM (up to 20 nm resolution [43]), will clarify further details of RyR network structure and its relationship with other networks formed, e.g., by mitochondria [44] or Ca^2+^ channels of different types and isoforms [45,46,47] directly or indirectly interacting with the RyR network. 

Future numerical modeling will include and test the functional importance of these relationships. The presence of spatial repulsion between CRUs suggests that there exists a molecular mechanism that actively maintains minimum spacing. Further biophysical studies will also clarify the nature of CRU repulsion and, possibly, coalescence [40]. Thus, the CRU sizes are likely determined, in part, by the repulsion/coalescence balance controlled by autonomic modulation to reach the maximal (or optimal) effect. 

The antibody mAb C33-3 used to detect RyRs in SANCs was generated against RyR2, but also cross-reacted with RyR1. However, using RT-PCR, it was reported that the sinoatrial node contained both RyR2 and RyR3, but not RyR1 [48]. The relative abundance of RyR2 and RyR3 was distinct between cells located at the periphery vs. those at the center of the sinoatrial node: RyR2 was higher at the periphery and RyR3 was higher in cells at the center of the node [49]. Thus, the present study did not investigate all RyRs, but was focused on cardiac type RyR2. Further studies are needed to clarify CRU composition and function with respect to both RyR2 and RyR3 isoforms in SANCs residing in different regions of the node.

### 4.5. Implications for Clinical and Aging Research

Disorder within biological systems tends to increase with aging across all scales. Thus, heterogeneity (i.e., disorder) in CRU sizes and spatial distribution is also expected to increase in older people, but as a result, CICR will actually be enhanced, thereby elevating the AP rate. This enhancement could partially compensate for the reduced intrinsic heart rate commonly associated with aging. However, the same increase in structural disorder of the RyR network would diminish β-adrenergic responsiveness, as demonstrated in this study, contributing to (and explaining in part) the decline in heart rate reserve observed in older people. 

As we mentioned in the Introduction, dysfunction of SANCs can lead to sick sinus syndrome, which is associated with a variety of life-threatening arrhythmias [3,4,5,6]. Electronic pacemakers are used to treat this condition, but impose lifestyle restriction and can cause severe side effects. To improve electronic pacemakers and create new approaches, for example, biological pacemakers [50], it is important to understand the natural origin of the heartbeat, which is “still mysterious after all these years” (cited from [23]). SANC operates via a coupled-clock mechanism [7], and its “Ca^2+^ clock” operates via synchronous and rhythmic release of Ca^2+^ via the RyR cluster network. Our new findings regarding the structure–function relation of the network can help direct the development of new prophylactic and therapeutic strategies. Specifically, our results suggest that healthy operation of the network requires a delicate balance of order and disorder in RyR cluster positions and sizes that determines and ensures the respective healthy balance of robustness and flexibility of cardiac pacemaking. Dysfunction of this balance could be a new factor in heart rate reserve decline in aging and disease. Our numerical model provides a new platform to investigate how molecular defects in CRUs and CRU network structure may disrupt pacemaking in diseases like sick sinus syndrome. Incorporating measured CRU network properties will help to provide mechanistic insights into pacemaker pathologies and develop new biological pacemakers.

### 4.6. Conclusions

CICR interplay of CRUs of various sizes and locations regulates and optimizes cardiac pacemaker cell operation under various physiological conditions and could be a key factor in heart rate reserve decline in aging and disease. Overall, our study elucidates how the specialized architecture of the CRU network in SANC regulates ignition and propagation of Ca^2+^ sparks to generate rhythmic AP firing. The combination of super-resolution imaging and computational modeling provides a powerful approach to link subcellular Ca^2+^ release patterns to whole-cell pacemaker activity.

## Figures and Tables

**Figure 1 cells-13-01885-f001:**
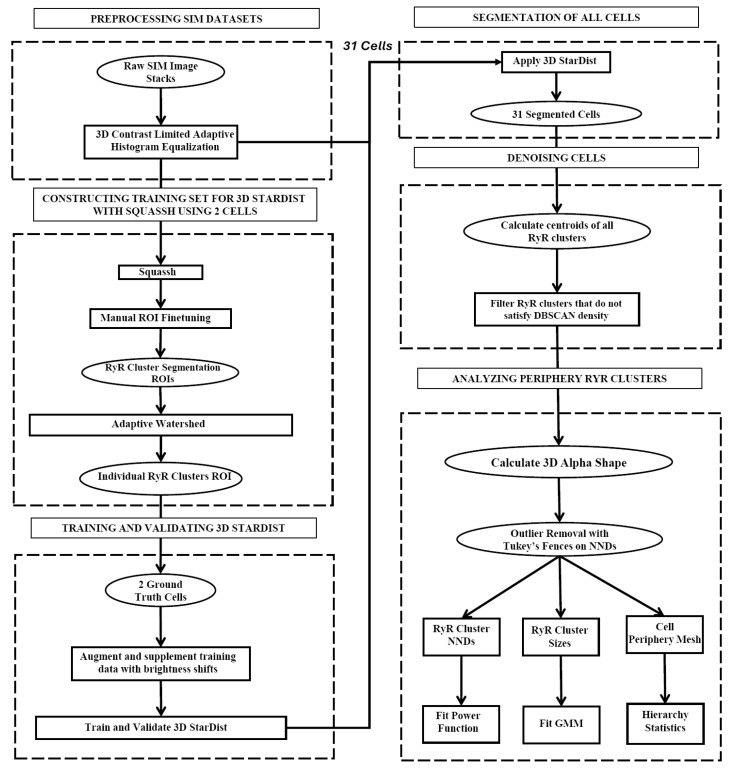
**Flowchart of the peripheral RyR cluster analysis algorithm.** Presented is our image-processing algorithm for the precise segmentation and analysis of peripheral RyR clusters in 3D SIM imaging data in SANCs. Initially, the data undergo cropping and contrast enhancement through 3D CLAHE. The main segmentation process is conducted using the 3D StarDist neural network, which is trained using ground truth data generated through the Squassh software and refined via adaptive watershed. Following segmentation, RyR clusters that are part of the cell are identified and extracted using the density-based spatial clustering of applications with noise (DBSCAN) algorithm, which detects high-density clusters within the dataset. The culmination of this process is the generation of a 3D alpha shape encapsulating the spatial distribution of the RyR clusters on the periphery of SANCs. The peripheral RyR cluster sizes, the distances, and their alpha shape mesh are exported for further statistical processing.

**Figure 2 cells-13-01885-f002:**
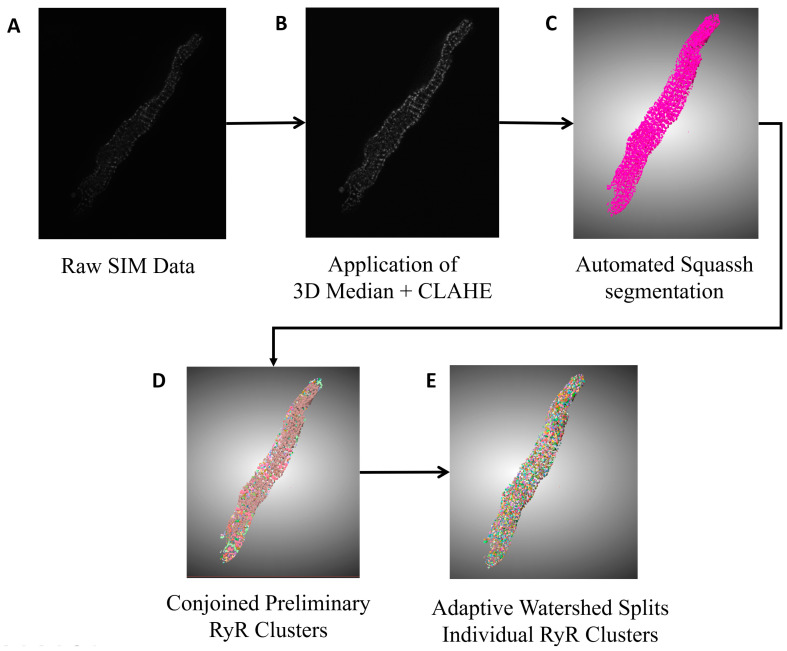
**Visual representation of transformations and segmentation of flowchart.** (**A**): The initial, unprocessed SIM image data (in 3 dimensions) of RyR immunofluorescence in SANC, serving as the primary source for image analysis. (**B**): The enhanced and denoised raw SIM data. (**C**): The preliminary segmentation from Squassh with a purple ROI. (**D**): The stage where distinct RyR clusters are isolated, establishing boundaries between adjoining regions using a 3D 26-connectivity strategy. (**E**): The Watershed algorithm’s role in further refining the segmentation, capable of isolating individual RyR clusters even in complex spatial arrangements.

**Figure 3 cells-13-01885-f003:**
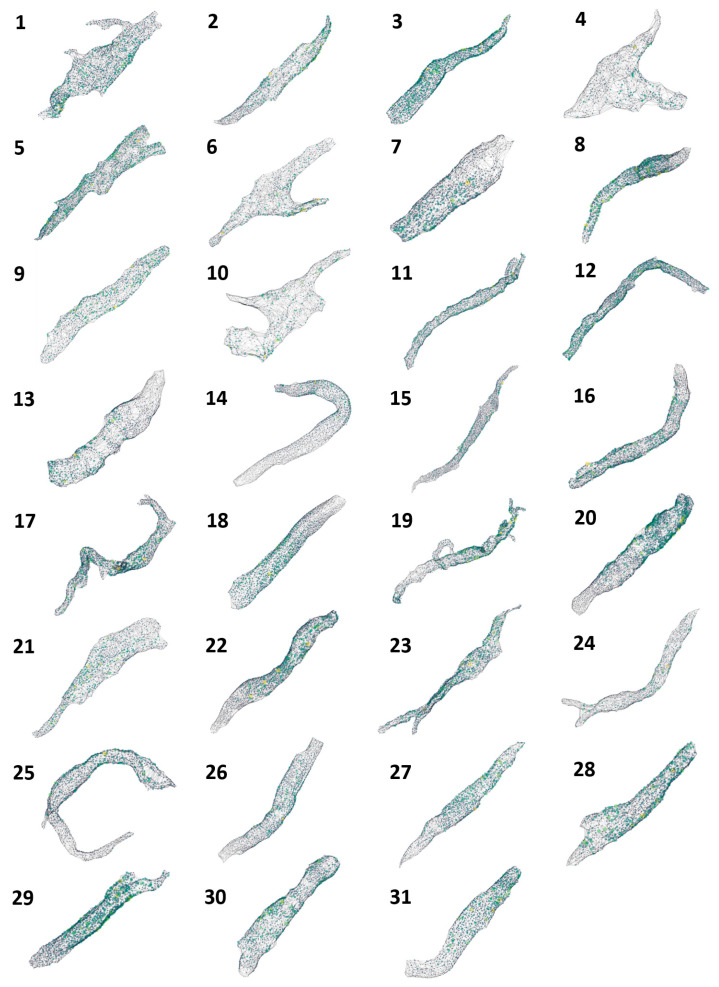
**Three-dimensional visualization of RyR cluster distribution in all 31 rabbit SANCs analyzed in the study**. Each cell is represented by a unique number (1–31) corresponding to the cell identifiers used in Table 1 and Table 2. The visualizations depict the spatial distribution of RyR clusters within each cell, with clusters represented as colored points. The color intensity indicates the relative sizes of the clusters, with brighter colors (yellow to green) representing larger clusters and dimmer colors (blue to gray) representing smaller clusters. These visualizations provide a comprehensive overview of the RyR cluster organization across the entire sample set, allowing for visual comparison of cluster distributions and densities among cells of different morphologies. Cells are shown at various zoom levels to closely fit the panel size. Three-dimensional representations of each cell with its scale bar are available in html format in GitHub https://github.com/alexmaltsev/SANC/tree/main/3D%20Visualizations (accessed 13 November 2024). Each html file can be opened in a web browser for a computer mouse-interactive view (rotation and zoom-in and -out).

**Figure 4 cells-13-01885-f004:**
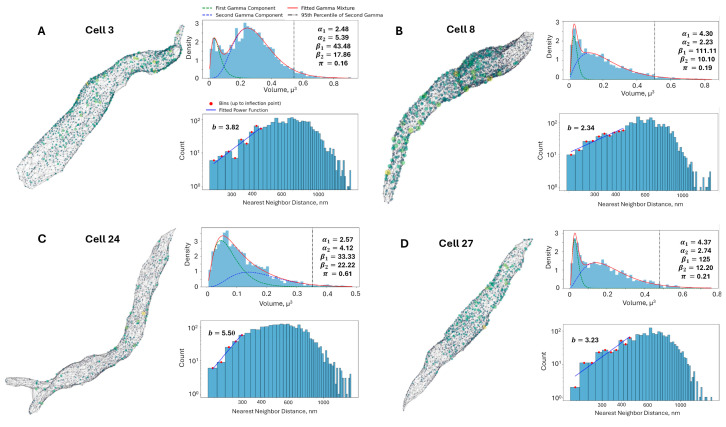
**Analysis of RyR cluster distribution and nearest-neighbor distances in representative rabbit SANC.** (**A**–**D**): Four representative cells (3, 8, 24, and 27 in Figure 3), each presenting three key analyses. First, a 3D visualization depicts RyR clusters as colored spheres scaled to their volumes (yellow for largest, purple for smallest) within a gray mesh representing the cell surface. Second, a fitted Gamma Mixture Model for RyR cluster size distributions is shown, with blue histograms representing observed data, green and blue dashed lines indicating individual gamma components, and a red line showing the overall fitted mixture. Model parameters (α_1_, α_2_, β_1_, β_2_, π) and the 95th percentile of the second gamma component (vertical black dash-dot line, marking large CRUs) are displayed. Third, a log-log histogram of RyR cluster nearest neighbor distances is presented, where blue bars show observed distances, red dots mark bins up to the inflection point used for power law fitting (blue line), and the fitted power law exponent (**B**) is provided. This comprehensive analysis reveals cell-to-cell variability in RyR cluster organization and spacing but overall consistency, offering insights into the spatial arrangement of Ca^2+^ release sites in SANC.

**Figure 5 cells-13-01885-f005:**
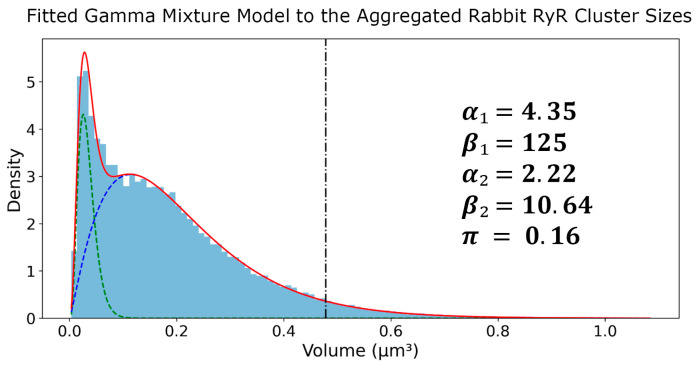
**Gamma mixture model analysis of RyR cluster sizes in rabbit SANCs.** Probability density function (PDF) of RyR cluster sizes derived from 31 rabbit SANCs, encompassing 70,982 clusters. The distribution is fitted with a two-component Gamma Mixture Model, with parameters (α_1_, β_1_, α_2_, β_2_, π) displayed. The blue histogram represents experimental data, while green and blue dashed lines show individual gamma components. The red line depicts the overall fitted mixture, and the vertical black dash–dot line marks the 95th percentile of the second gamma component (marking large CRUs).

**Figure 6 cells-13-01885-f006:**
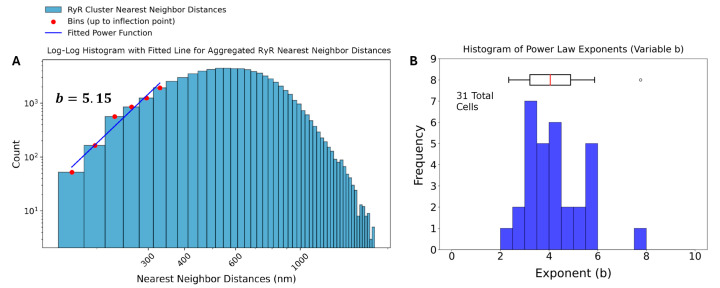
**Analysis of RyR cluster nearest-neighbor distances in rabbit sinoatrial node cells.** (**A**) Log-log histogram of nearest-neighbor distances (NND) for ryanodine receptor (RyR) clusters. The blue bars represent the observed NND distribution. Red dots indicate bins up to the inflection point, which were used to fit a power function (blue line). The fitted power law exponent b = 3.81 is displayed, suggesting strong repulsion among RyR clusters at short distances. (**B**) Histogram of power law exponents (b) derived from individual analysis of 31 cells, with an accompanying box plot.

**Figure 7 cells-13-01885-f007:**
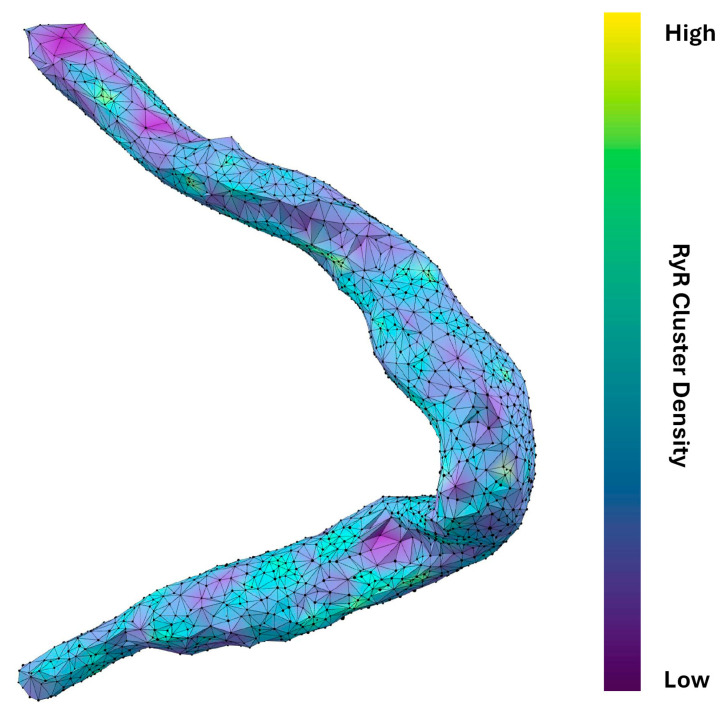
**Spatial distribution and connectivity analysis of RyR clusters in a rabbit SANC.** Detailed visualization of RyR cluster distribution in cell #14 (in Figure 3). The cell’s shape is outlined by a network of interconnected black vertices, each representing an RyR cluster. The surface coloring, transitioning from purple to yellow (via blue and green), indicates the density and connectivity of these clusters. Purple areas signify regions where clusters strongly repel each other, resulting in lower cluster density and fewer neighboring connections. Conversely, blue → green → yellow areas represent regions where this repulsion is weaker, leading to higher-density regions with more interconnected clusters. The varying degrees of repulsion between RyR clusters across different areas of the cell provide insights into the spatial organization of Ca^2+^ release sites, which influence CICR and AP firing rate. Specifically, this visualization technique effectively highlights potential hotspots of Ca^2+^ signaling activity within the cell that is likely associated with higher density of RyR clusters.

**Figure 8 cells-13-01885-f008:**
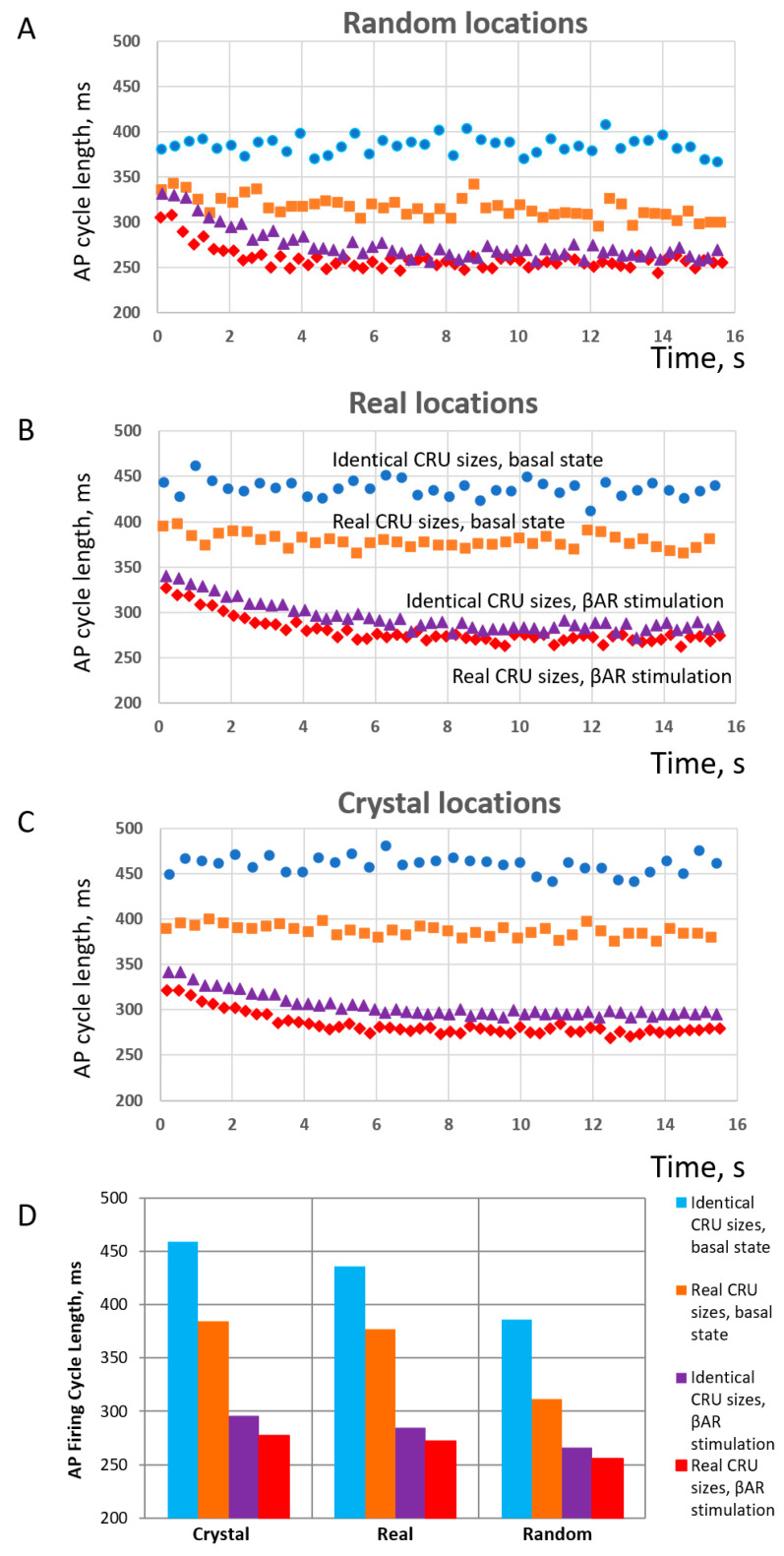
**Results of numerical model simulations of SANC function: heterogeneities in both CRU sizes and locations facilitate LCR propagation and increase AP firing rate in a cooperative manner but decrease the effect of βAR stimulation in terms of relative change in AP firing rate; at the same time, the presence of heterogeneities in both sizes and locations allows higher absolute AP firing rates to be reached during βAR stimulation.** (**A**–**C**): Intervalograms for AP cycle length (CL) in 12 numerical model simulations during 16 s for 6 models featuring different distributions of CRU sizes and distances in basal state and during βAR stimulation (shown by text labels in the panel (**B**)). (**D**): Graph showing average CL calculated for steady-state AP firing during 6 to 16 s of the simulations (see also Appendix A). Explanation for symbols: blue circles-identical CRU sizes in basal state; orange squares–real CRU sizes in basal state; purple triangles–identical CRU sizes in βAR stimulation; red diamonds–real CRU sizes in βAR stimulation.

**Figure 9 cells-13-01885-f009:**
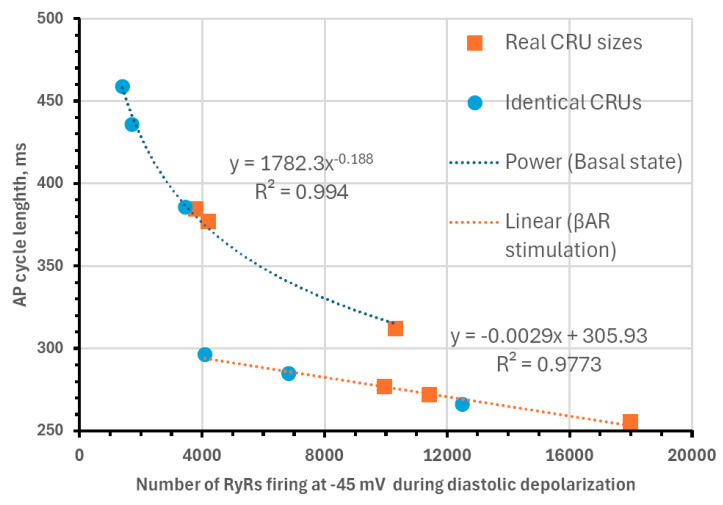
**Biophysical mechanism of how CRU distribution affects AP firing rate.** Shown is the relationship between of AP cycle length (CL) and the total number of RyRs in all firing CRUs at −45 mV, i.e., at the threshold of *I_CaL_* activation. Both in basal-state firing (fitted by power function, R^2^ = 0.994) and during βAR stimulation (fitted by linear function, R^2^ = 0.9773), CL was closely linked to the RyR firing at this critical timing within diastolic depolarization.

**Table 1 cells-13-01885-t001:** **Morphological and RyR cluster characteristics of rabbit sinoatrial node cells.** Detailed statistics for 31 rabbit sinoatrial node cells, including their surface area, volume, number of RyR clusters, and metrics related to RyR cluster distances and sizes. The data encompass a wide range of cellular morphologies and RyR cluster distributions. Notably, cells 3, 8, 14, 24, and 27 (in Figure 3) are included in this comprehensive dataset, allowing for direct comparison of their characteristics with the broader cell population.

	Cell Statistics			RyR Cluster Distances		RyR Cluster Sizes	
Cell #	Surface Area (μm²)	Volume (μm³)	#RYR Clusters	Mean (nm)	SD (nm)	Mean (μm³)	SD (μm³)
1	2494	2886	2775	586.78	191.25	0.21	0.17
2	2166	3050	1871	633.27	210.61	0.16	0.12
3	2698	5234	2035	722.19	218.38	0.26	0.15
4	2177	3695	1359	657.48	239.13	0.13	0.09
5	3182	4777	3341	579.93	212.91	0.17	0.17
6	2708	3868	1901	647.02	233.15	0.12	0.08
7	2435	6409	1566	695.58	243.85	0.19	0.14
8	1923	3119	2085	627.35	185.29	0.19	0.15
9	1583	1992	1573	617.39	209.54	0.19	0.16
10	2879	5088	1675	705.36	268.41	0.12	0.09
11	2300	2722	3046	555.98	174.95	0.20	0.18
12	3175	4822	3603	600.26	179.40	0.20	0.13
13	3097	8328	1976	702.53	229.66	0.14	0.09
14	2841	4579	2648	669.51	196.30	0.17	0.11
15	2129	2150	3243	530.22	164.79	0.16	0.13
16	2240	3431	2470	587.66	204.89	0.18	0.16
17	2139	2238	2577	570.93	185.03	0.19	0.16
18	2457	4984	2074	704.40	207.67	0.18	0.12
19	2688	2813	2575	606.69	196.25	0.17	0.15
20	2457	3735	2201	648.78	203.09	0.19	0.15
21	5256	15899	3364	736.34	253.06	0.20	0.14
22	2195	3898	2548	557.56	181.65	0.16	0.13
23	2318	2772	2292	633.41	194.55	0.23	0.18
24	2474	3655	2328	605.15	212.17	0.12	0.09
25	1983	2200	2205	596.14	191.90	0.18	0.14
26	2908	6224	2370	692.24	210.53	0.18	0.12
27	1877	3121	1639	676.19	209.62	0.18	0.14
28	1567	2278	1633	596.97	195.09	0.17	0.13
29	2564	4668	1987	694.89	212.81	0.19	0.12
30	3117	8181	2406	671.59	238.41	0.20	0.15
31	1640	2553	1483	687.51	198.09	0.22	0.14
AVERAGE	2505	4367	2285	638.62	208.14	0.18	0.13

**Table 2 cells-13-01885-t002:** **RyR cluster distribution analysis in rabbit SANCs.** Detailed analytical results for the 31 rabbit SANCs, focusing on key aspects of RyR cluster distribution: repulsion (b) and Gamma Mixture Model (GMM) parameters (α_1_, β_1_, α_2_, β_2_, π). The repulsion parameter b indicates the strength of spatial repulsion between RyR clusters. The GMM parameters describe the distribution of RyR cluster sizes. Notably, cells 3, 8, 14, 24, and 27 in previous figures are included in this dataset, allowing for a more comprehensive understanding of their RyR cluster characteristics in the context of the entire cell population. The aggregated results refer to the results that are calculated from all RyR clusters in all 31 cells in a single plot.

	Repulsion	Gamma Mixture Model (GMM)				
Cell #	b	$\alpha_{1}$	$\beta_{1}$	$\alpha_{2}$	$\beta_{2}$	π
1	2.59	4.20	142.86	2.43	9.52	0.19
2	5.87	4.05	142.86	2.28	12.99	0.12
3	3.82	2.48	43.48	5.39	17.86	0.16
4	2.51	4.79	111.11	2.42	16.67	0.17
5	7.76	4.51	142.86	2.12	8.55	0.34
6	4.65	5.20	125.00	2.77	20.83	0.13
7	3.01	3.47	100.00	3.04	13.51	0.20
8	2.34	4.30	111.11	2.23	10.10	0.19
9	4.05	3.93	125.00	2.19	9.17	0.23
10	3.13	5.05	111.11	2.62	18.87	0.18
11	3.79	4.57	142.86	1.99	7.87	0.22
12	5.76	2.03	35.71	3.36	14.29	0.23
13	4.2	4.45	125.00	2.99	19.61	0.10
14	4.07	1.94	12.99	8.03	35.71	0.74
15	5.71	4.38	111.11	2.18	11.90	0.14
16	5.12	4.67	142.86	1.80	8.13	0.20
17	3.21	4.18	125.00	2.14	9.17	0.22
18	3.9	1.65	10.53	8.60	34.48	0.69
19	3.81	4.15	111.11	2.10	9.90	0.21
20	3.2	4.55	142.86	2.24	10.42	0.12
21	4.1	4.21	111.11	2.99	12.66	0.18
22	5.35	3.64	100.00	2.21	11.63	0.22
23	3.96	3.86	90.91	2.60	9.43	0.18
24	5.5	2.57	33.33	4.12	22.22	0.61
25	3.21	8.62	333.33	1.91	9.71	0.12
26	5.74	1.83	11.90	7.22	29.41	0.72
27	3.23	4.37	125.00	2.74	12.20	0.21
28	4.45	2.69	55.56	2.79	12.66	0.30
29	4.59	1.89	11.24	9.14	37.04	0.69
30	4.19	3.96	125.00	2.89	11.76	0.22
31	3.03	1.52	8.26	8.11	29.41	0.62
AVERAGE	4.19	3.80	100.68	3.54	16.05	0.29
AGGREGATED	5.15	4.35	125.00	2.22	10.64	0.16

## Data Availability

All original data files were deposited to Harvard Dataverse at https://doi.org/10.7910/DVN/N0OLGI (accessed 13 November 2024). Code Availability: Code for image data analysis was uploaded to GitHub: https://github.com/alexmaltsev/SANC (accessed 13 November 2024), https://github.com/valventura/SANC (accessed 13 November 2024). Our new model of rabbit SANC with CRUs of various sizes written in Delphi programming language was uploaded to GitHub: https://github.com/victoramaltsev/CRU-based-SANC-model (accessed 13 November 2024).

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
