# Peer review of "Structure-Function Relationship of the Ryanodine Receptor Cluster Network in Sinoatrial Node Cells"

_cells, 2024, doi:10.3390/cells13221885_

Round 1
Reviewer 1 Report
Comments and Suggestions for Authors
The manuscript entitled "Structure-Function Relationship of the Ryanodine Receptor Cluster Network in Sinoatrial Node Cells" by Alexander V Maltsev et al explores the relationships between the distribution of ryanodine receptor (RyR) Ca2+ release channels and the spatiotemporal characteristics of Ca2+ signals in these cells. The distribution of RyR-dependent Ca2+ release units (CRUs) in rabbit sinoatrial node cells (SANC) was determined by super-resolution structured illumination microscopy. The influence of this distribution and the effects of beta-adrenergic stimulation on nodal action potential (AP) frequency were predicted by computer modelling. Key findings are that rabbit SANC contain two subpopulations of CRU and that this is predicted to influence the range of AP frequencies that these cells can generate under different conditions.
This work is of high quality, is well described and is likely to be of interest to a wide range of readers. It is also of biomedical significance, in terms of understanding dysregulation of SANC electrophysiology in "sick sinus" syndrome. However, the manuscript could be improved by taking the following points into consideration:
1) The antibody used to detect RyRs in SANC cells (mAb C33-3) was generated against RyR2, but also cross-reacts with RyR1. However, using RT-PCR, it was reported that rabbit SANC contain both RyR2 and RyR3, but not RyR1. Furthermore, the relative abundance of these channels is distinct between cells located at the periphery versus those at the centre of the node: RyR2 is higher at the periphery and RyR3 higher in cells at the centre of the node (Tellez et al, (2006) Circ Res. 99(12):1384-93). Consequently, it is unlikely that antibody mAb C33-3 would detect all of the RyRs (and so, CRUs) within a rabbit SANC. This Reviewer appreciates that it would be particularly onerous to repeat this study using either RyR-isoform selective antibodies (and that it would be technically difficult to draw quantitative comparisons using these), a genuinely pan-RyR antibody, or a fluorescent-ryanodine congener. Also, in a mixed population of RyR subtypes, many of the individual channel complexes are likely to be heterotetramers. However, this matter should be considered in Section 4.4, Limitations and future directions.
2) Figure 3 should include an indicator of scale (a scale-bar).
3) Figure that are in the main body of text should not be referred to as "supplementary".
4) There are two grammatical errors that are repeated throughout the text: i) "Ca" should be "Ca2+" (if the authors are referring to ionized calcium); and ii) "Ca releases" should be "Ca2+ release" or "Ca2+ release events". Please correct these errors.
Author Response
General Comment: The manuscript entitled "Structure-Function Relationship of the Ryanodine Receptor Cluster Network in Sinoatrial Node Cells" by Alexander V Maltsev et al explores the relationships between the distribution of ryanodine receptor (RyR) Ca2+ release channels and the spatiotemporal characteristics of Ca2+ signals in these cells. The distribution of RyR-dependent Ca2+ release units (CRUs) in rabbit sinoatrial node cells (SANC) was determined by super-resolution structured illumination microscopy. The influence of this distribution and the effects of beta-adrenergic stimulation on nodal action potential (AP) frequency were predicted by computer modelling. Key findings are that rabbit SANC contain two subpopulations of CRU and that this is predicted to influence the range of AP frequencies that these cells can generate under different conditions.
This work is of high quality, is well described and is likely to be of interest to a wide range of readers. It is also of biomedical significance, in terms of understanding dysregulation of SANC electrophysiology in "sick sinus" syndrome.
Response: Thank you for considering our work as of high quality and biomedical significance.
However, the manuscript could be improved by taking the following points into consideration:
Comment 1: 1) The antibody used to detect RyRs in SANC cells (mAb C33-3) was generated against RyR2, but also cross-reacts with RyR1. However, using RT-PCR, it was reported that rabbit SANC contain both RyR2 and RyR3, but not RyR1. Furthermore, the relative abundance of these channels is distinct between cells located at the periphery versus those at the centre of the node: RyR2 is higher at the periphery and RyR3 higher in cells at the centre of the node (Tellez et al, (2006) Circ Res. 99(12):1384-93). Consequently, it is unlikely that antibody mAb C33-3 would detect all of the RyRs (and so, CRUs) within a rabbit SANC. This Reviewer appreciates that it would be particularly onerous to repeat this study using either RyR-isoform selective antibodies (and that it would be technically difficult to draw quantitative comparisons using these), a genuinely pan-RyR antibody, or a fluorescent-ryanodine congener. Also, in a mixed population of RyR subtypes, many of the individual channel complexes are likely to be heterotetramers. However, this matter should be considered in Section 4.4, Limitations and future directions.
Response 1: We agree that specificity of antibodies is very important. As you requested, we addressed this matter in the revised Section 4.4, Limitations and future directions (new paragraph at the end of the section).
Comment 2: 2) Figure 3 should include an indicator of scale (a scale-bar).
Response 2: Figure 3 shows 3D views of each cell (not usual microscopic 2D views) at various zoom levels to closely fit the panel size. Yes, we agree, the scale bar is important, and we have added it, but for each cell 3D representation in html format uploaded in GitHub https://github.com/alexmaltsev/SANC/tree/main/3D%20Visualizations. This link was also added to the figure legend. So, the readers can now rotate and zoom-in-and-out each cell with its scale-bar attached (also in 3D) by computer mouse in a web browser.
Comment 3: 3) Figure that are in the main body of text should not be referred to as "supplementary".
Response 3: We have revised the Figure 3 legend to exclude “supplementary”.
Comment 4: 4) There are two grammatical errors that are repeated throughout the text: i) "Ca" should be "Ca2+" (if the authors are referring to ionized calcium); and ii) "Ca releases" should be "Ca2+ release" or "Ca2+ release events". Please correct these errors.
Response 4: We changed Ca to Ca2+ and we also replaced Ca releases by Ca2+ release events or by Ca2+ release. We carefully read the text several times and fixed other gramma errors that we found.
Reviewer 2 Report
Comments and Suggestions for Authors
This paper offers an insightful exploration of the structural and functional relationship of Ryanodine Receptor 2 (RyR2) cluster networks in sinoatrial node cells (SANC), with particular focus on their role in heart rate reserve decline associated with aging and disease. By employing super-resolution structured illumination microscopy (SIM), the authors achieve a highly detailed 3D visualization of calcium release unit (CRU) networks, which illuminates the influence of CRU size and spatial organization on signal propagation within the sinoatrial node.
The study's findings underscore the critical role of CRU distribution in optimizing calcium-induced calcium release (CICR) efficiency, paving the way for targeted interventions aimed at restoring optimal network functionality in the context of aging and cardiac diseases. These results hold promising implications for the development of novel therapeutic strategies to enhance heart rate reserve and mitigate age-related cardiac dysfunction.
Author Response
General Comment 1: This paper offers an insightful exploration of the structural and functional relationship of Ryanodine Receptor 2 (RyR2) cluster networks in sinoatrial node cells (SANC), with particular focus on their role in heart rate reserve decline associated with aging and disease. By employing super-resolution structured illumination microscopy (SIM), the authors achieve a highly detailed 3D visualization of calcium release unit (CRU) networks, which illuminates the influence of CRU size and spatial organization on signal propagation within the sinoatrial node.
Response: Thank you for considering our paper as an insightful exploration of the structural and functional relationship of Ryanodine Receptor 2 (RyR2) cluster networks.
General Comment 2: The study's findings underscore the critical role of CRU distribution in optimizing calcium-induced calcium release (CICR) efficiency, paving the way for targeted interventions aimed at restoring optimal network functionality in the context of aging and cardiac diseases. These results hold promising implications for the development of novel therapeutic strategies to enhance heart rate reserve and mitigate age-related cardiac dysfunction.
Response: Thank you for considering our results as holding promising implications for the development of novel therapeutic strategies to enhance heart rate reserve and mitigate age-related cardiac dysfunction.
Reviewer 3 Report
Comments and Suggestions for Authors
The study is well-designed and presented as it examines the geometrical and spatial CPU networks in rabbit sinoatrial node cells using experimental as well as computational methods. A few suggestions for the authors:
1. 384 words is too long for an abstract. please stick to a 250 word limit and only mention key information.
2. Labels showing Figure X on each figure must be deleted.
3. "Figure 8. Results of numerical model simulations:" - Please change the figure title to mention key finding(s). Also, there is no need for each word to start with a capital letter (e.g., Figure 1, 3... etc.).
4. Please define all abbreviations at first mention even if they are standard in the subfield. (e.g., action potential [AP]).
5. "(as previously described)" - delete brackets.
6. The Discussion could be improved by mentioning the clinical implications of the findings. This is very important to better contextualize the new data with what is already known.
7. Also discuss how the new findings may help direct the development of new prophylactic and therapeutic strategies.
Author Response
General Comment: The study is well-designed and presented as it examines the geometrical and spatial CPU networks in rabbit sinoatrial node cells using experimental as well as computational methods.
Response: Thank you for considering our work as well-designed and presented.
A few suggestions for the authors:
Comment 1: 1. 384 words is too long for an abstract. please stick to a 250 word limit and only mention key information.
Response 1: We have reduced the abstract to 250 words
Comment 2: 2. Labels showing Figure X on each figure must be deleted.
Response 1: We have deleted labels showing Figure X in the manuscript file we submit now
Comment 3: 3. "Figure 8. Results of numerical model simulations:" - Please change the figure title to mention key finding(s). Also, there is no need for each word to start with a capital letter (e.g., Figure 1, 3... etc.).
Response 3: We have revised the title of Figure 8 legend to reflect key findings.
Comment 4: 4. Please define all abbreviations at first mention even if they are standard in the subfield. (e.g., action potential [AP]).
Response 4: We have defined abbreviation at first mention. We also have a list of all abbreviations at the end of the manuscript (before References).
Comment 5: 5. "(as previously described)" - delete brackets.
Response 5: fixed.
Comment 6: 6. The Discussion could be improved by mentioning the clinical implications of the findings. This is very important to better contextualize the new data with what is already known.
Response 6: We wrote a new section 4.5. Implications for clinical and aging research, in which we discuss clinical implications of the findings.
Comment 7: 7. Also discuss how the new findings may help direct the development of new prophylactic and therapeutic strategies.
Response 7: In the new section 4.5. we also discuss how the new findings may help direct the development of new prophylactic and therapeutic strategies
Round 2
Reviewer 3 Report
Comments and Suggestions for Authors
The authors have adequately addressed my comments.